# Multi-omics prediction of immune-related adverse events during checkpoint immunotherapy

Ying Jing[1,9], Jin Liu[2,9], Youqiong Ye [1], Lei Pan[3], Hui Deng[3], Yushu Wang[1], Yang Yang[4], Lixia Diao [4], Steven H. Lin [5], Gordon B. Mills [6], Guanglei Zhuang [2,7✉], Xinying Xue [3✉] & Leng Han [1,8✉]

Immune-related adverse events (irAEs), caused by anti-PD-1/PD-L1 antibodies, can lead to fulminant and even fatal consequences and thus require early detection and aggressive management. However, a comprehensive approach to identify biomarkers of irAE is lacking. Here, we utilize a strategy that combines pharmacovigilance data and omics data, and evaluate associations between multi-omics factors and irAE reporting odds ratio across different cancer types. We identify a bivariate regression model of LCP1 and ADPGK that can accurately predict irAE. We further validate LCP1 and ADPGK as biomarkers in an independent patient-level cohort. Our approach provides a method for identifying potential biomarkers of irAE in cancer immunotherapy using both pharmacovigilance data and multi-omics data.

[1] Department of Biochemistry and Molecular Biology, The University of Texas Health Science Center at Houston McGovern Medical School, Houston, TX 77030, USA. [2] State Key Laboratory of Oncogenes and Related Genes, Shanghai Cancer Institute, Ren Ji Hospital, School of Medicine, Shanghai Jiao Tong University, 200217 Shanghai, China. [3] Department of Respiratory and Critical Care, Beijing Shijitan Hospital, Capital Medical University; Peking University Ninth School of Clinical Medicine, 100038 Beijing, China. [4] Department of Bioinformatics and Computational Biology, The University of Texas MD Anderson Cancer Center, Houston, TX 77030, USA. [5] Department of Radiation Oncology, Division of Radiation Oncology, The University of Texas MD Anderson Cancer Center, Houston, TX 77030, USA. [6] Department of Cell, Development, and Cancer Biology, Knight Cancer Institute, Oregon Health and Science University, Portland, OR 97239, USA. [7] Shanghai Key Laboratory of Gynecologic Oncology, Ren Ji Hospital, School of Medicine, Shanghai Jiao Tong University, 200217 Shanghai, China. [8] Center for Precision Health, The University of Texas Health Science Center at Houston, Houston, TX 77030, USA. [9] These authors contributed equally: Ying Jing, Jin Liu. ✉email: zhuangguanglei@gmail.com; xinyingxue2010@163.com; leng.han@uth.tmc.edu

I mmune-related adverse events (irAEs) during anti-programmed death 1 (PD-1) or anti-programmed death ligand 1 (PD-L1) antibody therapy, resulting from immune activation combined with disturbed immunologic homeostasis, can affect any organ systems and in some cases can be lethal[1]. Pneumonitis, the most common fatal irAE, results in 10% death rate, and accounts for 35% of anti-PD-1/PD-L1-related fatalities[2]. Myocarditis, the most lethal irAE, causes ~50% mortality[3]. Therefore, the predictive biomarkers of irAEs are required to determine the benefit/risk ratio for patients receiving anti-PD-1/PD-L1 therapy. T-cell receptor (TCR) diversity[4], CD8+ T-cell clonal expansion[5], and tumor mutational burden (TMB)[6] have been suggested to potentially predict irAE albeit on the basis of a single factor or conducted in a limited number of cases. Therefore, a comprehensive approach to identify biomarkers of irAE is lacking. In particular, it is challenging to obtain a patient sample cohort with enough sample size, and the traditional approach may require multiple years of multi-center efforts.

In this study, we investigate potential predictors for irAE risk in patients receiving anti-PD-1/PD-L1 therapies across 26 tumor types by integrating real-world pharmacovigilance and molecular omics data. We identify the bivariate linear-regression model of LCP1 and ADPGK that can accurately predict irAE, and validate LCP1 + ADPGK model predictive potential in an independent patient cohort. Our approach provides a method for identifying potential biomarkers of irAE in cancer immunotherapy.

## Results

### Analysis of known factors in predicting irAE.
To identify potential biomarkers of irAE in anti-PD-1/PD-L1 therapy, we retrieved from the US Food and Drug Administration Adverse Event Reporting System (FAERS) a total of 52,282 adverse events (AEs) from 18,706 patients for 26 different cancer types receiving anti-PD-1/PD-L1 therapy. Among these patients, 3706 (19.8%) had at least one irAE. We calculated the irAE reporting odds ratio (ROR) by comparing the proportion of reporting irAEs for anti-PD-1/PD-L1 agents with the proportion of reporting irAEs for all other drugs in the database[7]. IrAE ROR varied by tumor type and the highest irAE ROR observed for lung adenocarcinoma (LUAD) (3.29, 95% confidence interval (CI), 2.97–3.65), while the lowest value was observed for uterine carcinosarcoma (UCS) (0.65, 95% CI, 0.02–4.18) (Fig. 1a and Supplementary Table 1). We collected six factors related to irAEs, including TMB[6], T-cell receptor (TCR) diversity[8], interferon (IFN) α level[9], tumor necrosis factor (TNF) α level[9], eosinophils[10], and neutrophils[11]. Interestingly, these factors are also biomarkers of immune therapy response based on positive associations between the incidence of irAEs and benefit for patients treated with immune checkpoint inhibitors[12,13]. Indeed, we observed a marginally significant correlation between irAE ROR and objective response rates[14] in anti-PD-1/PD-L1 therapy ($Rs = 0.44$; $P = 0.049$; Supplementary Fig. 1). We further collected 36 factors related to immune therapy response, including TMB[15], cytolytic activity[16], and neoantigen load[15]. We evaluated the association of these factors calculated from molecular data of The Cancer Genome Atlas (TCGA) and irAE risks based on the individual safety reports from FAERS. We identified seven potential predictors, including cytolytic activity (Spearman R, $Rs = 0.64$; false discover rate (FDR) = 0.01), IFN γ signature ($Rs = 0.61$, FDR = 0.01), PD-1 expression ($Rs = 0.60$, FDR = 0.01), TCR diversity ($Rs = 0.59$, FDR = 0.01), macrophages M1 ($Rs = 0.55$, FDR = 0.03), CD8+ T-cell abundance ($Rs = 0.50$, FDR = 0.05), naive B cells ($Rs = 0.49$, FDR = 0.05) (Fig. 1b, Supplementary Fig. 2, and Supplementary Table 2). To identify more powerful predictive models, we combined these seven factors and evaluated the performance of bivariate models

by Spearman correlation and goodness of fit by the log-likelihood ratio test[17]. The combination of CD8+ T-cell abundance with TCR diversity or naive B cells achieved significantly improved goodness of fit of models compared to using the single factors (Fig. 1c and Supplementary Fig. 3). In particular, the combination of CD8+ T cells and TCR diversity achieved maximum predictive efficacy ($Rs = 0.75$, FDR = $8.24 \times 10^{-4}$) (Fig. 1d and Supplementary Table 3). The correlation coefficient ($Rs$, 0.75) suggested that 56% ($Rs^2$, 0.56) of observed irAE ROR was explained by this bivariate regression model. We assessed the multicollinearity of these seven factors by the variance inflation factor[18,19], and observed no multicollinearity for TCR diversity and CD8+ T cells (Supplementary Fig. 4). We also found that TCR diversity and CD8+ T-cell abundance exhibited no significant correlation ($P = 0.26$), suggesting the independent prediction of irAE. We further evaluated the performance of the combinations of other factors with TCR diversity—CD8+ T-cell abundance bivariate model. No trivariate models achieved higher correlation coefficients or increased accuracy (Supplementary Table 4).

### Comprehensive identification for potential irAE biomarkers.
We further sought to identify additional predictive factors for irAE by conducting a comprehensive screening across mRNA, miRNA, lncRNA and protein expression, and nonsilent gene mutations across 26 cancer types. The majority of the top hits that resulted were gene expressions (Supplementary Table 5), which were highly enriched in immune response processes, including T-cell activation and cell killing (Fig. 2a). This provided further support for the T cells as the pivotal regulators in irAEs. Of particular interest, the lymphocyte cytosolic protein 1 (LCP1), which is involved in T-cell activation[20], achieved the highest correlation coefficient ($Rs = 0.82$, FDR = $6.69 \times 10^{-3}$, Fig. 2b). Combinations between any two of the top ten significant irAE correlated genes (Supplementary Fig. 5) suggested that the combination of LCP1 with most of the other genes achieved better predictive performance (Fig. 2c, Supplementary Fig. 6, and Supplementary Table 6). Among these, adding the adenosine diphosphate dependent glucokinase (ADPGK), which is mediating metabolic shift during T-cell activation[21], to LCP1 led to a linear-regression model with the best accuracy among all the bivariate models ($Rs = 0.91$, FDR = $7.94 \times 10^{-9}$, Fig. 2d and Supplementary Table 6). We evaluated the multicollinearity of these top ten genes by the variance inflation factor[18,19] and observed no multicollinearity for LCP1 and ADPGK (Supplementary Fig. 7). Combinations of the third gene did not improve the predictive value of the LCP1 and ADPGK bivariate model (Supplementary Table 7). We further screened the combination of significant factors and genes to identify more powerful combinations, and did not discover models have better performance (Supplementary Figs. 8 and 9, and Supplementary Table 8). Considering the straightforward estimation of LCP1 and ADPGK, this combination maybe easier to translate into clinical practice. As far as we know, no study reported that LCP1 and/or ADPGK is associated with immunotherapy response yet. We further performed correlation analysis between objective response rate[14] and LCP1/ADPGK, and observed no significant associations (Supplementary Fig. 10), suggesting limited confounding effect from efficacy, at least for LCP1 and ADPGK.

### Validation of LCP1 and ADPGK as irAE biomarkers.
To validate the predictive power of LCP1 and ADPGK, we further collected a validation cohort of 28 cancer patients receiving anti-PD-1/PD-L1 inhibitors with both high-quality formalin-fixed paraffin-embedded (FFPE) pre-treatment tumor tissues and clinicopathological information (Supplementary Table 9 and

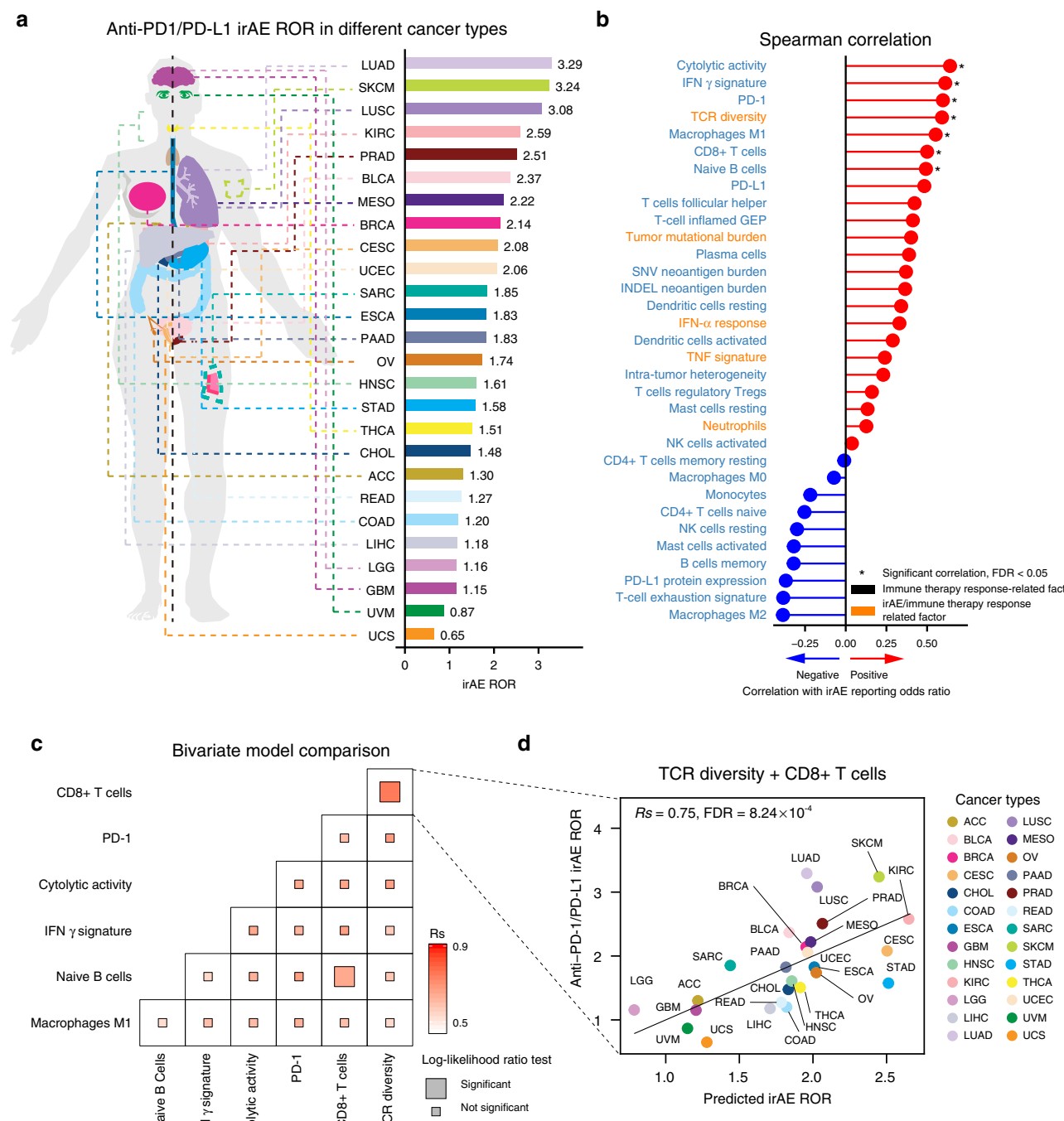

**Fig. 1 Evaluation of the association between irAE and related factors. a** Anatomic sites of cancer types (left panel), and irAE ROR across 26 cancer types (right panel). **b** Spearman correlation between irAE ROR and 36 factors for positive correlation (red lollipop) and negative correlation (blue lollipop). * indicates significant correlation (FDR < 0.05); cytolytic activity FDR = 0.01, Interferon (IFN) γ signature FDR = 0.01, PD-1 FDR = 0.01, T-cell receptor (TCR) diversity FDR = 0.01, macrophages M1 FDR = 0.03, CD8+ T cells FDR = 0.05, naive B cells FDR = 0.05; irAE and immune therapy response-related factors are marked in orange. **c** Comparison of performance of bivariate models in predicting irAE for all combinations of six significantly correlated variables. Spearman R (*Rs*) was calculated between predicted and observed irAE ROR. The shade of the square indicates the *Rs*, and the size indicates the significance of the log-likelihood ratio test. **d** Combined effect of TCR diversity and CD8+ T-cell bivariate model (Spearman correlation, *Rs* = 0.75, FDR = $8.24 \times 10^{-4}$). The equation of the bivariate model is 0.31 × TCR diversity + 8.87 × CD8+ T cells + 0.27. irAE immune-related adverse events, ROR reporting odds ratio, FDR false discovery rate, LUAD lung adenocarcinoma, SKCM skin cutaneous melanoma, LUSC lung squamous cell carcinoma, KIRC kidney renal clear cell carcinoma, PRAD prostate adenocarcinoma, BLCA bladder urothelial carcinoma, MESO mesothelioma, BRCA breast invasive carcinoma, CESC cervical squamous cell carcinoma and endocervical adenocarcinoma, UCEC uterine corpus endometrial carcinoma, SARC sarcoma, ESCA esophageal carcinoma, PAAD pancreatic adenocarcinoma, OV ovarian serous cystadenocarcinoma, HNSC head and neck squamous cell carcinoma, STAD stomach adenocarcinoma, THCA thyroid carcinoma, CHOL cholangiocarcinoma, ACC adrenocortical carcinoma, READ rectum adenocarcinoma, COAD colon adenocarcinoma, LIHC liver hepatocellular carcinoma, LGG brain lower-grade glioma, GBM glioblastoma multiforme, UVM uveal melanoma, UCS uterine carcinosarcoma.

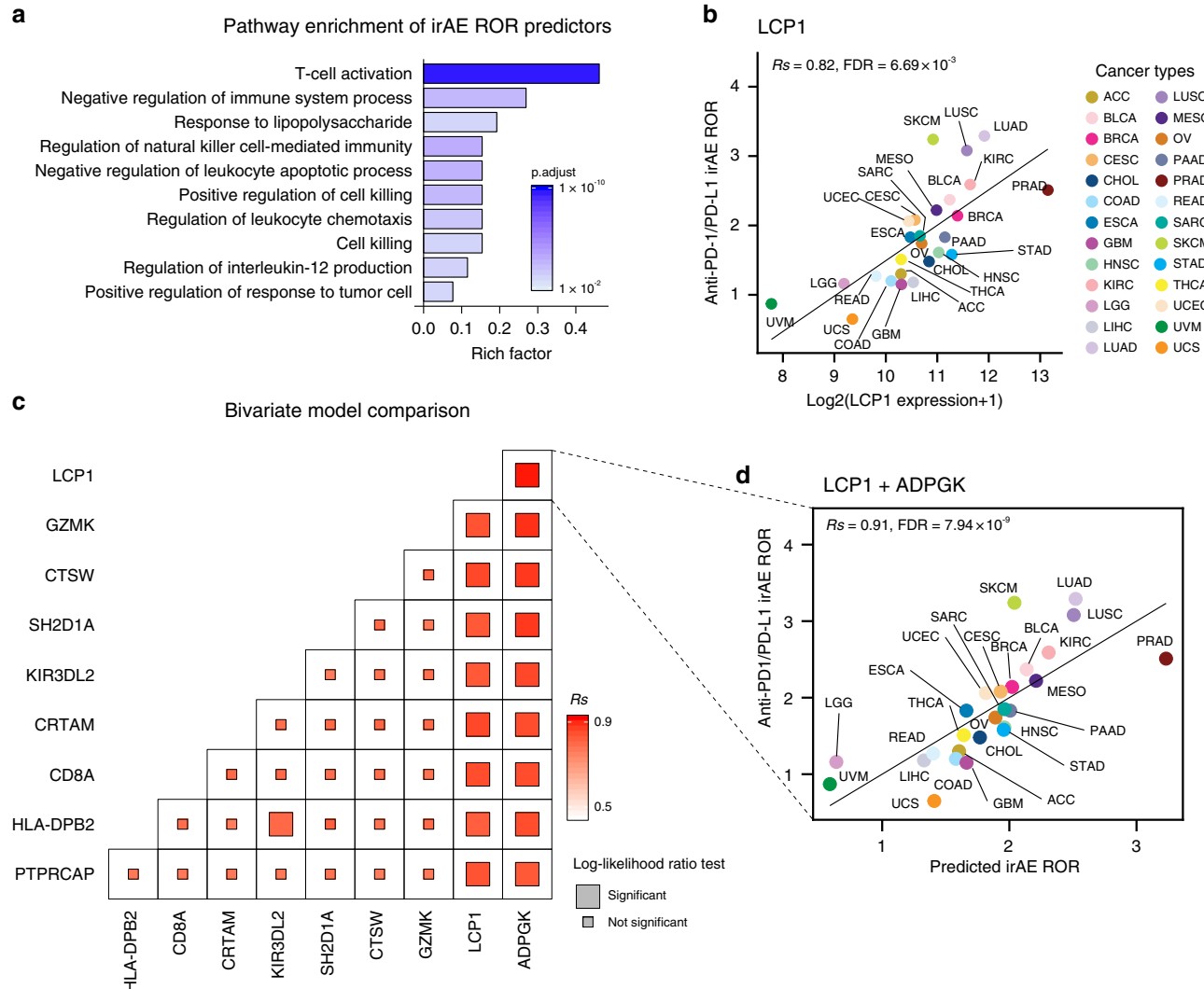

**Fig. 2 Comprehensive identification of potential irAE predictors. a** Pathway enrichment of the top ten genes significantly correlated with irAE ROR across multiple cancer types. **b** Spearman correlation between LCP1 expression and irAE ROR. **c** Comparison of performance of bivariate models in predicting irAE for all combinations of the top ten irAE ROR significantly correlated genes. Spearman correlation (*Rs*) was calculated between the predicted and observed irAE ROR. The shade of the square indicates the *Rs*, and the size indicates the significance of the log-likelihood ratio test. **d** Combined effect of LCP1 and ADPGK bivariate model (Spearman correlation, *Rs* = 0.91, FDR = $7.94 \times 10^{-9}$). The equation of the bivariate regression model is 0.37 × LCP1 + 0.70 × ADPGK – 9.10.

Supplementary Fig. 11). The median age of the patients was 56 years (range, 37–82 years), with 22 (78.6%) male patients and 6 (21.4%) female patients. In all, 26 of 28 (92.9%) patients were diagnosed as lung cancer. The expression level of LCP1 and ADPGK were assessed by immunohistochemistry in our validation cohort. LCP1 and ADPGK have stronger staining in the irAE group (Fig. 3a). We quantified the immunostaining signals for the protein expression of LCP1 and ADPGK using the Aperio ImageScope software v14.3 with Positive Pixel Count v9 (PPCv9) algorithm. Consistently, LCP1 (*P* value = 0.008) and ADPGK (*P* value = 0.010) were higher in patients with irAEs when compared with patients without irAEs (Fig. 3b). The geometric mean of LCP1 and ADPGK was also higher in pre-treatment tumor samples of patients with irAEs (*P* value = 0.005, Fig. 3c). The area under the receiver-operating characteristic curve (AUC) of LCP1 and ADPGK to predict irAE was 0.78 and 0.78, while the combination of LCP1 and ADPGK had a better AUC value as 0.80 (Fig. 3d). Furthermore, LCP1, ADPGK, and LCP1+ADPGK successfully predicted pneumonitis in 26 lung cancer patients with AUC as 0.74, 0.76, and 0.77, respectively (Supplementary

Fig. 12), suggesting the potential utility of LCP1 and ADPGK in predicting a specific type of irAE in a specific cancer. We further tested the CD8 predictive potential in our validation cohort. We did not observe significantly increased CD8 IHC staining signals in pre-treatment tumors in the irAE group. In addition, we found the AUC value of CD8 is not comparable to LCP1 and ADPGK (Supplementary Fig. 13), suggesting that the confounding effect of CD8 in irAE prediction is limited. Taken together, this independent patient-level cohort validated the predictive power of LCP1 and ADPGK for irAEs in cancer patients receiving anti-PD-1/PD-L1 inhibitors.

## Discussion
In this study, we systematically investigated potential predictors for irAE risk in patients receiving anti-PD-1/PD-L1 therapies across 26 tumor types by integrating real-world and molecular data. We identified seven potential predictors, and the combination of CD8+ T cells and TCR diversity achieved higher accuracy of prediction of irAE and decreased the unexplained

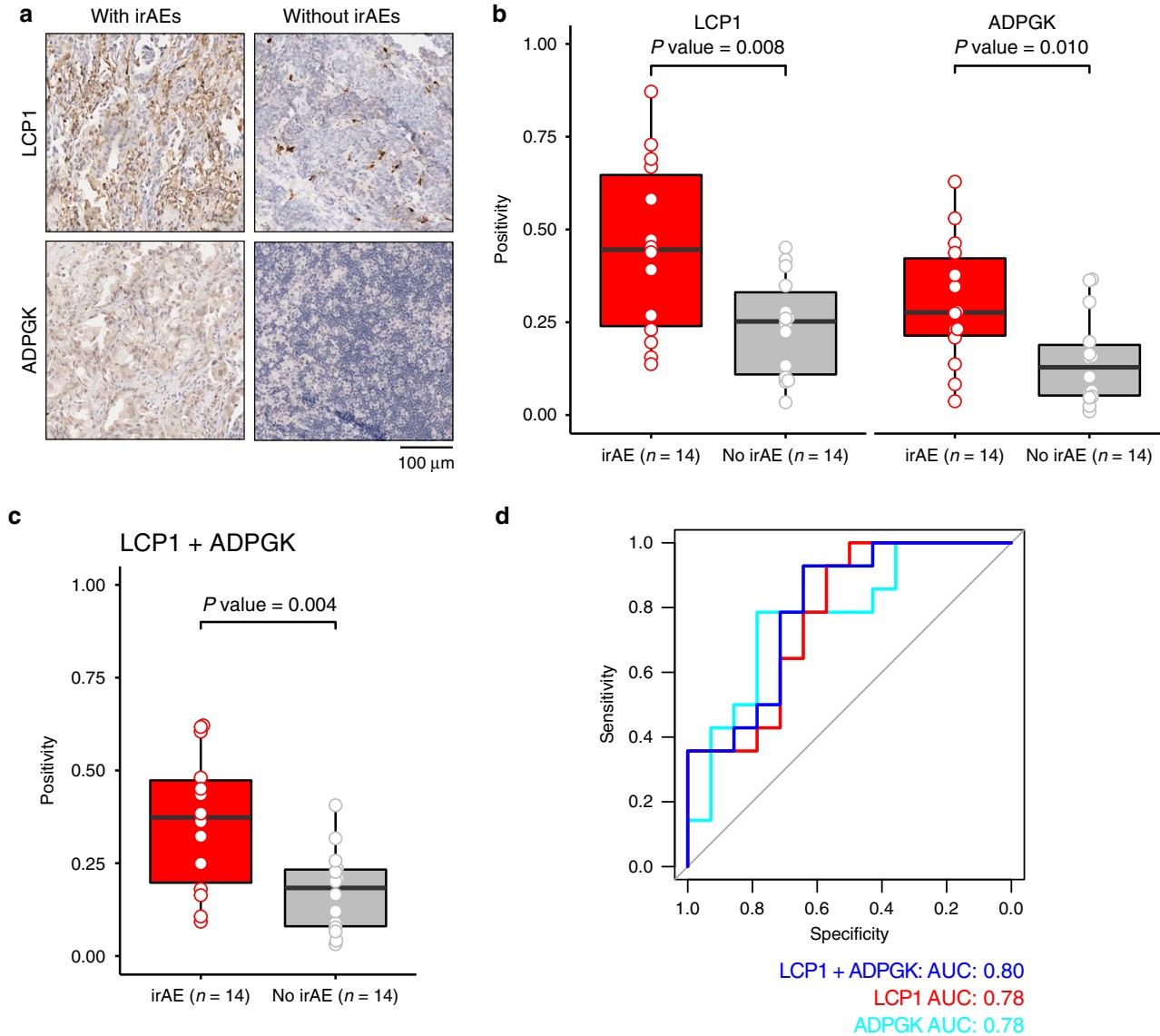

**Fig. 3 Validation of the predictive power of LCP1 and ADPGK in a patient cohort. a** Representative images of patients with irAEs and without irAEs through immunohistochemical (IHC) staining with antibodies against ADPGK and LCP1. Image size: $379 \times 379 \ \mu m^2$; scale bar: 100 μm. **b** Quantification of LCP1 and ADPGK IHC staining signals. Comparison between patients with irAE or without irAE is conducted by unpaired two-sided Student's t test. **c** Geometric mean of LCP1 and ADPGK staining signals. A comparison between patients with irAE and without irAE is conducted by unpaired two-sided Student's t test. **d** ROC curve of the LCP1, ADPGK, LCP1 + ADPGK in the validation cohort ($n = 28$). The boxes indicate the median ± 1 quartile, with whiskers extending from the hinge to the smallest or largest value within 1.5 interquartile range from the box boundaries. ROC receiver-operating characteristic, AUC area under the receiver-operating characteristic curve.

variance from 0.59 ($1-0.64^2$) to 0.44 ($1-0.75^2$). Considering that the unexplained variance was still 0.44, we further performed a large-scale comprehensive screening to identify better performing irAE ROR predictors. We identified potential irAE predictors that are enriched in the function of T-cell activation. The linear-regression model by the combination of *LCP1* and *ADPGK*, two genes related to T-cell activation[20,21], further decreased the unexplained variance from 0.44 ($1-0.75^2$) to 0.17 ($1-0.91^2$). Our results align well with the emergence of evidence of the involvement of T-cell activation in irAEs. Furthermore, the AUC value of LCP1 and ADPGK in our patient-level validation cohort achieved 0.8, suggesting that the combination of LCP1 and AGDPGK holds promise as biomarkers for irAEs. Meanwhile, our proof-of-concept study set up an analytic framework to explore irAE biomarkers and propose possible underpinning key factors, which might have important implications for the management of patients with immunotherapy. Our analysis enables the study of promising signals of immune-related toxicities in large sample cohorts, while collecting both molecular data and irAE information of thousands of patients with immunotherapy requires the collaboration of multicenter with several years' efforts. Future work is necessary to further study the predictive performance of LCP1 and ADPGK in larger anti-PD-1/PD-L1 patient cohorts. We envision that LCP1 and ADPGK might enable a pre-risk-check of patients before receiving anti-PD-1/PD-L1 agents with further study.

Nevertheless, our study has a limitation with a limited sample size. Ideally, it will be promising to obtain a large number of patient samples with or without irAEs, and then perform the multi-omics data to identify biomarkers. However, it is challenging to obtain a patient-sample cohort with enough sample size because the percentage of clearly defined of a certain type of irAE,

e.g., pneumonitis, is <5%[22,23]. It may require multiple years of multicenter efforts. Therefore, we utilize an alternative strategy to combine the power of real-world data and omics data. A similar strategy is also utilized to identify the potential biomarkers[6,14,24], suggesting this strategy is robust and powerful, especially in the absence of a large number of patient samples. Further studies with a larger sample size are necessary to comprehensively identify irAE.

## Methods

**Data analysis of individual safety report from FAERS**. We retrieved individual safety reports from FAERS [https://fis.fda.gov/sense/app/d10be6bb-494e-4cd2-82e4-0135608ddc13/sheet/7a47a261-d58b-4203-a8aa-6d3021737452/state/analysis] from July 1, 2014 to June 30, 2019. We collected only AE reports from anti-PD-1 agents (nivolumab, pembrolizumab, cemiplimab) and anti-PD-L1 (atezolizumab, avelumab, durvalumab) suspected of causing AEs. We excluded those cases also treated with anti-CTLA-4 agents (ipilimumab, tremelimumab). We used the AE terms in peer-reviewed irAE management guidelines[23] to define irAEs. We performed disproportionality analysis[7] to assess the risk of irAEs via calculating the ROR by using the full database as the comparator. Patients were categorized to irAE group when they have one type of irAEs[23].

**Data analysis from TCGA and independent datasets**. Molecular data, including mRNA expression, miRNA expression, protein expression, and somatic mutations across 26 cancer types were downloaded from TCGA data portal [https://portal.gdc.cancer.gov/]. TCR diversity, neoantigen load, estimated immune cell abundance, and intratumor heterogeneity were downloaded from the GDC PanImmune Data Portal [https://gdc.cancer.gov/about-data/publications/panimmune][25]. TMB was calculated by the number of nonsilent somatic mutations per sample[26]. We used GSVA R package[27] v1.3 to compute the T-cell-inflamed gene expression profiling (GEP) level[28] in each sample based on the T-cell-inflamed GEP signature from Ayers et al.[29]. Cytolytic activity was calculated as the geometric mean of the gene expression of two cytolytic markers (GZMA and PRF1)[16]. IFN γ signature was obtained from Ayers et al.[29].

**Identification of biomarkers by combining omics data and real-world data**. In our analysis, the number of cancer types is far less than variables (26 cancer types with >50,000 variables, including ~20,000 mRNA expression, ~12,000 noncoding RNA expression, ~18,000 gene mutations, ~200 protein expression, and ~2400 miRNA expression), which may result in an inflated type-I-error and subsequently introduce more false positives[30–35] if employ other advanced algorithms, e.g., Lasso, Elastic net, and Ridge. Therefore, we adopted an approach as described in a previous study[17] that they evaluated the correlation between single variables and response rate, and then added variables to obtain bivariate models to achieve better performance. The median values of each factor were calculated for each cancer type. The anatomic illustration was generated by R package gganatogram[36,37] v1.1. We performed leave-one-out cross-validation in predicting irAE ROR from bivariate and trivariate linear-regression models using the R package caret v6.0. The predictive performance was estimated using the Spearman rank correlation coefficient ($Rs$) and unexplained variance ($1 - Rs^2$). The goodness of fit of the models was compared by the log-likelihood ratio test using the R package lmtest v0.9. For the bivariate model fitness comparison, the log-likelihood ratio test was performed between two-factor models and the single-factor model with the highest $Rs$. For the trivariate model fitness comparison, the log-likelihood ratio test was performed between the trivariate model and the bivariate model. We used variance inflation factor to assess multicollinearity by vif function of the car R package v3.0. Pathway enrichment was conducted using the R package clusterprofiler[38] v3.14. The calculation of the ROC curve was completed by pROC R package v1.16. Multiple comparisons were Benjamini–Hochberg adjusted by p. adjust function of the base R language, version 3.5.0. Statistical significance was defined as two-sided $P < 0.05$ and/or FDR < 0.05.

**Immunohistochemistry in our patient cohort**. The study was conducted in accordance with ethical guidelines of U.S. Common Rule, and was approved by the Ethics Committee of Beijing Shijitan Hospital. Written informed consent was obtained from all patients. We performed a retrospective review of cancer patients with lung cancer, gastrointestinal (GI) cancer, genitourinary (GU) cancer, or other cancers receiving anti-PD-1/PD-L1 treatment from 2017 to 2019 in Beijing Shijitan Hospital. To identify high-confidence irAE in patients, we used relatively stringent criteria: (1) we only include patients with CT confirmed pneumonitis; (2) we only include the pneumonitis that requires and responds to steroids, immunosuppressants, or endocrine therapies; (3) two investigators independently determine the pneumonitis as immunologic etiology. Patients with incomplete demographic and follow-up information, receiving anti-PD-1/PD-L1 in other hospitals, or a history of anti-CTLA-4 therapy were excluded. Next, we kept patients who have archived high-quality pre-treatment FFPE tumor samples before receiving anti-PD-1/PD-L1 and any other treatment. In addition, the tumor tissue should contain

<30% necrosis. We collected a comparable number of anti-PD-1/PD-L1-treated cancer patients without any overserved irAEs with matched cancer types, stage, age, sex, and therapy of irAE group (Supplementary Fig. 11). Detailed clinical characteristics of the patient cohort were described in Supplementary Table 9. Immunohistochemistry (IHC) was performed on 5-μm-thick FFPE tumor tissue sections. Slides stained with primary antibodies against LCP1 (1:200, Cell Signaling Technology #3588), ADPGK (1:900, Novus Biologicals #NBP1-91653), and CD8 (working solution, MXB Biotechnology #RMA-0514). Slides were then washed and incubated with horseradish peroxidase-conjugated secondary antibodies (1:200, NeoBioscience, # ANR02-1). Immunoperoxidase staining was developed using the DAB system according to the manufacturer's instructions (Dako). Slides were counterstained with hematoxylin, dehydrated, and coverslipped using a mounting solution. Whole slides were scanned with an Aperio ScanScope system (Leica Biosystems) and quantified using the Aperio ImageScope software v14.3 with Positive Pixel Count v9 (PPCv9) algorithm. Areas of necrosis or artifacts in the slides were ignored. IHC signals were enumerated in seven random 20× fields, and averaged signals were used for each slide.

**Reporting summary**. Further information on research design is available in the Nature Research Reporting Summary linked to this article.

## Data availability

Individual safety records were downloaded from FAERS Public Dashboard [https://fis.fda.gov/sense/app/d10be6bb-494e-4cd2-82e4-0135608ddc13/sheet/7a47a261-d58b-4203-a8aa-6d3021737452/state/analysis]. The TCGA data were downloaded from TCGA data portal [https://portal.gdc.cancer.gov/] and GDC PanImmune Data Portal [https://gdc.cancer.gov/about-data/publications/panimmune]. All the remaining data are available within the Article, Supplementary Information files, or available from the author upon reasonable request. Source data are provided with this paper.

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

## Acknowledgements

This work was supported by the Cancer Prevention and Research Institute of Texas (grant nos. RR150085 and RP190570) to CPRIT Scholar in Cancer Research (L.H.). We thank LeeAnn Chastain for editorial assistance.

## Author contributions

L.H. conceived and supervised the project. Y.J., G.Z., X.X., and L.H. designed and performed the research. Y.J., J.L., Y.Y., Y.W., and L.D. performed data analysis. J.L., L.P., H.D., G.Z., and X.X. collected the validation cohort and performed the IHC staining. Y.J., S.H.L., G.B.M., and L.H. interpreted the results. Y.J., G.B.M., G.Z., and L.H. wrote the paper.

## Competing interests

G.B.M. has sponsored research support from AstraZeneca, Critical Outcomes Technology, Karus, Illumina, Immunomet, Nanostring, Tarveda, and Immunomet, and is on the Scientific Advisory Board for AstraZeneca, Critical Outcomes Technology, ImmunoMet, Ionis, Nuevolution, Symphogen, and Tarveda. The remaining authors declare no competing interests.
