## [Peer Review File · Nature Communications]

Reviewers' comments:

Reviewer #1 (Remarks to the Author): Immunogenomics

In this work, Jing and colleagues leverage existing multi-omics data from multiple tumor types to identify potential predictors of irAEs in immunotherapy-treated cancer patients.

While the work is very interesting, below are three critical points that I feel need to be addressed for this manuscript/work to be useful to the field in its present form:

1) The irAE reporting data are largely confounded by efficacy - tumor types that are more actively and widely treated, with greater response rates, are more likely to develop irAEs, this could cause overlap of the signal (linearity) with response rates. This reduces the utility of the 'predictor' since at this point response and irAE rate cannot be uncoupled. Likewise, many of the identified 'markers' explaining variability support this, as they are 'activated T cells, etc' - the same markers of response.

2) There is no single-sample prediction. At this point knowing the 'ROR' of an irAE shows no demonstration of being able to predict an irAE within a tumor type, on the individual basis.

3) There is no consideration of host factors, which according to existing literature, are more likely to be contributors to irAEs. Tumor tissue-specific data, are exactly that, tumor specific. There are some data that certain toxicities occur more frequently in certain tumor types when treatment patterns are consistent between the tumor types (e.g. pneumonitis in lung cancer), but even this observation doesn't necessarily point toward a tumor-specific pathogenesis, rather more likely from the existing inflammatory patterns in the lung.

Minor: What about patients receiving concurrent anti-CTLA-4? Are these culled out of the existing data? For instance are the authors including ANY patient treated with anti-PD-1 (including those with dual ICB?)

Reviewer #2 (Remarks to the Author): Biostatistical analysis

This paper presents multi-dimensional prediction models of immune-related AEs and response during immunotherapy. While prediction model building is the key component of this paper, the used prediction methods are quite naïve. Following are specific comments.

1. Although the title says the prediction models are multi-dimensional, in fact the fitted prediction models are only bi-dimensional. With many candidate predictors, it is not clear why the authors propose to use only two predictors for prediction of irAE since the predictive power possibly will be increasing by including more predictors.

2. Page 4 reports FDRs for different predictors. There are different methods to calculate FDR, e.g. Benjamini-Hochberg method and Storey's q-value method. It's not clear how the FDRs were calculated.

3. Fig 1c and 1d, Fig 2c and 2d: The authors (1) identify top 10 predictors based on the univariate association with irAE, and (2) develop prediction models for irAE using two highly correlated predictors among the 10 predictors. This is a poor approach because (1) variable selection based on

univariate association does not guarantee a high predictive power for the final multi-dimensional prediction model, and (2) having correlated predictors will reduce the predictive power because of the multi-collinearity.

Re: Manuscript ID: NCOMMS-20-10390-T

Reviewers' comments:

Reviewer #1 (Remarks to the Author): Immunogenomics

In this work, Jing and colleagues leverage existing multi-omics data from multiple tumor types to identify potential predictors of irAEs in immunotherapy-treated cancer patients. While the work is very interesting, below are three critical points that I feel need to be addressed for this manuscript/work to be useful to the field in its present form:

1) The irAE reporting data are largely confounded by efficacy - tumor types that are more actively and widely treated, with greater response rates, are more likely to develop irAEs, this could cause overlap of the signal (linearity) with response rates. This reduces the utility of the 'predictor' since at this point response and irAE rate cannot be uncoupled. Likewise, many of the identified 'markers' explaining variability support this, as they are 'activated T cells, etc' - the same markers of response.

Response: We thank the reviewer for appreciating the interesting of our work, as well as those valuable comments.

Immune related adverse events (irAEs) occur as a result of immune activation against host normal tissues or organs, and it remains unclear whether irAEs indicates concurrent antitumor immune response¹⁻³. Several studies reported positive association between irAE and PFS or OS for urothelial cancer⁴, lung cancer⁵ and melanoma patients⁶, while other studies showed no association between irAE and PFS or OS^{7,8}. In addition, most irAEs reported during anti-PD-1/PD-L1 therapy develop within the first few weeks of treatment⁹⁻¹¹, suggesting further study should be performed to investigate the relationship between irAE rate and response rate. Meanwhile, we have performed a preliminary analysis of objective response rate and irAE ROR, which showed that only 19.4% (R_s^2 , 0.44²) of observed irAE ROR was explained by objective response rate (Supplementary Figure 1). This result suggests that the confounding effect by efficacy is limited in our analytical setting, and it is possible to distinguish irAE and response.

Indeed, our analysis indicated the T cell activation could play important roles in the irAE development of cancer patients. T cell activation is also considered as major factors in other process, including development of autoimmunity, kill infected bacterial or virus, and anti-cancer immune response¹². Despite their functions in multiple processes, it is still necessary to understand

their roles in each process. Nevertheless, we identified LCPI and ADPGK as potential irAE predictive biomarkers. As far as we know, no study reported that LCPI and/or ADPGK is associated with immunotherapy response yet. We further performed correlation analysis between objective response rate¹³ and LCPI/ADPGK, and observed no significant associations (Figure 1a-b). Taken together, our results suggested limited confounding effect from efficacy, at least for LCPI and ADPGK. We have added the results as Supplementary Fig. 11 and discussed in revised manuscript page 6, line 7-11.

Figure 1. Spearman correlation between anti-PD-1/PD-L1 objective response rate and LCPI (a) and ADPGK (b).

2) There is no single-sample prediction. At this point knowing the 'ROR' of an irAE shows no demonstration of being able to predict an irAE within a tumor type, on the individual basis.

Response: We thank the reviewer for the comments.

Ideally, it will be promising to obtain large number of patient samples with or without irAEs, and then perform the multi-omics data to identify biomarkers. However, it is challenging to obtain a patient sample cohort with enough sample size. According to the personal communication with the experts in the field, this will require multiple-year of multi-center efforts. Therefore, we utilize an alternative strategy that to combine the power of real-world data and omics data as we presented in our original submission. Similar strategy is also utilized to identify the potential biomarkers¹⁴⁻¹⁷, suggesting this strategy is robust and powerful, especially in the absence of large number of patient samples. We have discussed in the revised manuscript (page 8 line 10-16).

Furthermore, we totally agree that the patient level data will be necessary to this study. Therefore, we tried our best to collect biospecimens of cancer patients treated with anti-PD-1/PD-L1 therapies

to validate our analysis. We hope the reviewer will appreciate the difficulty to obtain the individual patient samples with or without irAEs, especially at the pandemic of COVID-19. We collected formalin-fixed paraffin-embedded (FFPE) biospecimens from pathologically confirmed cancer patients before receiving anti-PD-1/PD-L1 antibodies. The cohort was selected based on the availability of clinical pathologic information and high-quality FFPE archive. We included 28 patients for the validation, 14 patients with irAEs and 14 patients without irAEs. Expression level of LCP1 and ADPGK were assessed by immunohistochemistry in our validation cohort. LCP1 and ADPGK has stronger staining in irAE group (Figure 2a). We quantified the immunostaining signals for the protein expression of LCP1 and ADPGK using the Aperio ImageScope software v14.3 with Positive Pixel Count v9 (PPCv9) algorithm. Consistently, LCP1 (student *t*-test, p-value = 0.008) and ADPGK (student *t*-test, p-value = 0.010) were upregulated in patients with irAEs when compared with patients without irAEs (Figure 2b). The geometric mean of LCP1 and ADPGK was also higher in patients with irAEs (p-value = 0.005, Figure 2c). The AUC of LCP1, ADPGK and LCP1+ADPGK were 0.78 (95%CI 0.60-0.95), 0.78 (95%CI 0.60-0.96), 0.80 (95%CI 0.63-0.97) (Figure 2d). This independent patient-level cohort validated our prediction, and we added the results in the revised manuscript (Results section: page 6-7; Online Methods section: page 10-11) and figures (Revised Figure 3).

Figure 2. Validation the predictive power of LCP1 and ADPGK in a patient cohort.

- a. Representative images of patients with irAEs and without irAEs through immunohistochemical (IHC) staining with antibodies against ADPGK and LCP1. Image size: 200×200 μm^2 .
- b. Quantification of LCP1 and ADPGK IHC staining signals. Comparison between patients with irAE or without irAE is conducted by unpaired two-sided student's *t*-test.
- c. Geometric mean of LCP1 and ADPGK staining signals. Comparison between patients with irAE and without irAE is conducted by unpaired two-sided student's *t*-test.
- d. ROC curve of the LCP1, ADPGK, LCP1+ADPGK in the validation cohort (n=28).

3) There is no consideration of host factors, which according to existing literature, are more likely to be contributors to irAEs. Tumor tissue-specific data, are exactly that, tumor specific. There are some data that certain toxicities occur more frequently in certain tumor types when treatment patterns are consistent between the tumor types (e.g. pneumonitis in lung cancer), but even this observation doesn't necessarily point toward a tumor-specific pathogenesis, rather more likely from the existing inflammatory patterns in the lung.

Response: We totally agree with the reviewer that pneumonitis happens more in lung cancer patients treated with anti-PD-1⁹. In a systematic review of PD-1 ICI trials, patients with melanoma had a higher frequency of gastrointestinal and skin irAEs¹⁸, not only skin. Each cancer type probably have their own specific bias of irAEs¹⁸. We are trying to avoid such issue in our analysis that patients were categorized to irAE group when they have one type of irAEs as summarized in a recent review⁹ (furtherly clarified in revised manuscript page 9, line 4-5). The similar approach has been done in a previous study¹⁶. Of course, the large number of samples within each cancer type will be better to address this issue, but such data is absent. Alternatively, we validated our prediction biomarkers with patient level data in lung cancer patients with pneumonitis. Please note, that 26 out of 28 patients from our validation cohort is lung cancer patients. Consistently, LCP1 (student *t*-test, p-value = 0.026), ADPGK (student *t*-test, p-value = 0.022) and geometric mean of LCP1 and ADPGK (student *t*-test, p-value = 0.015) were upregulated in patients with irAEs when compared with patients without irAEs (Figure 3a-b). The AUC of LCP1, ADPGK LCP+ADPGK were 0.74 (95%CI, 0.54-0.94), 0.76 (95%CI 0.56-0.95), 0.77 (95%CI 0.59-0.96) (Figure 3c). In summary, these results suggest that LCP1 and ADPGK have strong predictive power in prediction of irAEs in cancer, and it is also useful to predict one specific type of irAE in one cancer type. We

have added the results as Supplementary Fig. 12 and described in the revised manuscript page 7 line 4-7.

Figure 3. Validation the predictive power of LCP1 and ADPGK in lung cancer patients.

a. Quantification of LCP1 and ADPGK immunohistochemical (IHC) staining signals. Comparison between patients with irAE and without irAE is conducted by unpaired two-sided student's *t*-test.

b. Geometric mean of LCP1+ADPGK staining signals. Comparison between patients with irAE and without irAE is conducted by unpaired two-sided student's *t*-test.

c. ROC curve of the LCP1, ADPGK, LCP1+ADPGK in the validation cohort (n=26).

Minor: What about patients receiving concurrent anti-CTLA-4? Are these culled out of the existing data? For instance are the authors including ANY patient treated with anti-PD-1 (including those with dual ICB?)

Response: Our study focused on anti-PD-1/PD-L1 inhibitors. Therefore, we only considered safety reports for which the anti-PD-1/PD-L1 agents, nivolumab, pembrolizumab, atezolizumab, avelumab, durvalumab and cemiplimab were the suspected cause of adverse events. Retrieved cases received anti-CTLA-4 agents, ipilimumab and tremelimumab, were excluded. We employed

the same method as described in previous studies¹⁹⁻²², especially for anti-PD-1/PD-L1 agents¹⁶, which is quite standard to analyze real-world data. We clarified this in the revised manuscript (page 9 line 1-2).

Reviewer #2 (Remarks to the Author): Biostatistical analysis

This paper presents multi-dimensional prediction models of immune-related AEs and response during immunotherapy. While prediction model building is the key component of this paper, the used prediction methods are quite naïve. Following are specific comments.

1. Although the title says the prediction models are multi-dimensional, in fact the fitted prediction models are only bi-dimensional. With many candidate predictors, it is not clear why the authors propose to use only two predictors for prediction of irAE since the predictive power possibly will be increasing by including more predictors.

Response: We thank the reviewer for this valuable comment.

In our study, the multi-dimensional in the title means that we employed multi-dimensional omics data, including mRNA expression, miRNA expression, protein expression and gene mutation, not the model. We changed our title to “multi-omics prediction of immune-related adverse events during checkpoint immunotherapy” to avoid potential misunderstanding (Revise manuscript page 1 line 1).

We totally agreed with the reviewer that the predictive power possibly will be increasing by including more predictors. Our omics data are high dimensional but only contain 26 cancer types, it is challenging to use typical variable selection methods because of potential error^{23,24}. Therefore, we employed the similar approach from a previous study (JAMA Oncology, 2019)¹⁵ by adding variables to find models with better performance. Please also see response to reviewer 2 comment 3 where we explained in more details. In our original submission, we performed additional analysis to add other genes to LCP1+ADPGK bivariate model. Unfortunately, none of the trivariate model achieve better performance than the bivariate model (Figure 4). We have added the results as Supplementary Table 7 and described in revised manuscript (page 5, line 23 – page 6, line 1).

Figure 4. Combined effects of trivariate models for all combinations of top 10 genes and LCP1+ADPGK bivariate model.

2. Page 4 reports FDRs for different predictors. There are different methods to calculate FDR, e.g. Benjamini-Hochberg method and Storey's q-value method. It's not clear how the FDRs were calculated.

Response: We used the Benjamini-Hochberg method to adjust all of the statistical significance in our multiple tests. We clarified this in the revised manuscript (page 10 line 14-15).

3. Fig 1c and 1d, Fig 2c and 2d: The authors (1) identify top 10 predictors based on the univariate association with irAE, and (2) develop prediction models for irAE using two highly correlated predictors among the 10 predictors. This is a poor approach because (1) variable selection based on univariate association does not guarantee a high predictive power for the final multi-dimensional prediction model, and (2) having correlated predictors will reduce the predictive power because of the multi-collinearity.

Response: We thank the reviewer to point two potential issues:

(1) Variable selection.

We totally agreed with the reviewer that variable selection based on univariate association does not guarantee a high predictive power for the final multi-dimensional prediction model. We employed the similar approach from a previous study (JAMA Oncology, 2019)¹⁵. They evaluated

the correlation between single variables and response rate, and then added variables to obtain bivariate models which has better performance. They successfully obtain a trivariate model comprised of tumor mutational burden, CD8 T Cell and fraction of PD-1 to predict immunotherapy response rate. This is an alternative approach to generate computational model for the scenario that number of samples are far less than variables, i.e., $n < p$ (26 cancer types with >50,000 variables, including ~20,000 mRNA expression, ~12,000 noncoding RNA expression, ~18,000 gene mutations, ~200 protein expression, and ~2,400 miRNA expression). In this situation, other advanced algorithms, e.g., Lasso, Elastic net, and Ridge, may have an inflated type-I-error that may introduce more false positives, especially when tuning parameters²³⁻²⁸. We have added these descriptions in our revised manuscript (Online Methods section: page 9 line 20 – page 10 line 2). Furthermore, in our study, we aimed to use the computational approach to identify the potential biomarkers, instead of pursuing a model with the highest predictive power. Therefore, a patient-level validation maybe more important and robust than refining the model. In this situation, we collected 28 patient samples to validate that LCP1 and ADPGK are really good biomarkers for irAE (please also refer to our response to reviewer 1, comment 3).

(2) Multi-collinearity.

First, our original solution is more robust to the multi-collinearity issue as each of the 10 genes were selected univariately. Our bivariate model also alleviated this concern mostly as each model has only two genes. On the contrary, more advanced and complicated methods, e.g., Lasso, is more vulnerable to the multicollinearity issues²⁶. This is also one of the reasons we chose relatively simple model in our original submission.

Second, we assessed multicollinearity of the top 10 predictors by using the variance inflation factor (VIF)^{19,29}. Multi-collinearity is not observed for TCR diversity and CD8+ T cells (Figure 5a), LCP1 and ADGPK (Figure 5b) as the VIF is less than 4.0^{19,29}, suggesting no or limited effect of multicollinearity for TCR diversity and CD8+ T cells, LCP1 and ADPGK. We have added the multicollinearity evaluation in our revised manuscript (Results section: page 4 line 22 - page 5 line 1, page 5 line 21 - 23; Online Methods section: page 10 line 11 - 12) and figures (Revised Supplementary Figure 4 & 7).

Figure 5. Variance inflation factor (VIF) value of irAE ROR related factors and genes.

a. VIF value of seven significant factors associated with irAE ROR.

b. VIF value of top 10 factors associated with irAE ROR.

Third, we agree with the reviewer that multi-collinearity may cause some problems to predictive models in statistical inference. Specifically, the multicollinearity may only affect the coefficient estimation and interpretation of the individual variables^{30,31}. In our study, we aimed to identify the potential molecular biomarkers as a result of feature selection, instead of accurately estimating the coefficient value of individual variables. In this situation, our major conclusion that LCP1 and ADPGK may serve as the potential biomarker will not be affected. Furthermore, our independent patient-level cohort validated the predictive power of LCP1 and ADPGK.

Taken together, these results suggest the limited effect of multi-collinearity on our conclusion.

Reference

1. Saleh, K., Khalife-Saleh, N. & Kourie, H. R. Do immune-related adverse events correlate with response to immune checkpoint inhibitors? *Immunotherapy* 11, 257–259 (2019).
2. Havel, J. J., Chowell, D. & Chan, T. A. The evolving landscape of biomarkers for checkpoint inhibitor immunotherapy. *Nat. Rev. Cancer* 19, 133–150 (2019).
3. Pauken, K. E., Dougan, M., Rose, N. R., Lichtman, A. H. & Sharpe, A. H. Adverse Events Following Cancer Immunotherapy: Obstacles and Opportunities. *Trends Immunol.* 40, 511–523 (2019).
4. Maher, V. E. et al. Analysis of the Association Between Adverse Events and Outcome in Patients Receiving a Programmed Death Protein 1 or Programmed Death Ligand 1

- Antibody. *J. Clin. Oncol.* JCO.19.00318 (2019). doi:10.1200/jco.19.00318
5. Remon, J., Reguart, N., Auclin, E. & Besse, B. Immune-Related Adverse Events and Outcomes in Patients with Advanced Non–Small Cell Lung Cancer: A Predictive Marker of Efficacy? *J. Thorac. Oncol.* 14, 963–967 (2019).
 6. Eggermont, A. M. M. et al. Association between Immune-Related Adverse Events and Recurrence-Free Survival among Patients with Stage III Melanoma Randomized to Receive Pembrolizumab or Placebo: A Secondary Analysis of a Randomized Clinical Trial. *JAMA Oncol.* Jan 2, (2020).
 7. Horvat, T. Z. et al. Immune-related adverse events, need for systemic immunosuppression, and effects on survival and time to treatment failure in patients with melanoma treated with ipilimumab at memorial sloan kettering cancer center. *J. Clin. Oncol.* 33, 3193–3198 (2015).
 8. Weber, J. S. et al. Safety Profile of Nivolumab Monotherapy : A Pooled Analysis of Patients With Advanced Melanoma. *J. Clin. Oncol.* 35, 785–92 (2017).
 9. Martins, F. et al. Adverse effects of immune-checkpoint inhibitors: epidemiology, management and surveillance. *Nat. Rev. Clin. Oncol.* 16, 563–80 (2019).
 10. Ramos-, M. et al. Immune- related adverse events of checkpoint inhibitors. *Nat. Rev. Dis. Prim.* 6, 1–21 (2020).
 11. Kennedy, L. B. & Salama, A. K. S. A review of cancer immunotherapy toxicity. *CA. Cancer J. Clin.* 0, 1–19 (2020).
 12. Mills, K. H. G. TLR-dependent T cell activation in autoimmunity. *Nat. Rev. Immunol.* 11, 807–822 (2011).
 13. Yarchoan, M., Hopkins, A. & Jaffee, E. M. Tumor mutational burden and response rate to PD-1 inhibition. *N. Engl. J. Med.* 377, 2500–2501 (2017).
 14. Yarchoan, M., Hopkins, A. & Jaffee, E. M. Tumor mutational burden and response rate to PD-1 inhibition. *N. Engl. J. Med.* 377, 2500–2501 (2017).
 15. Lee, J. S. & Ruppin, E. Multiomics Prediction of Response Rates to Therapies to Inhibit Programmed Cell Death 1 and Programmed Cell Death 1 Ligand 1. *JAMA Oncol.* 5, 1614–1618 (2019).
 16. Bomze, D., Hasan Ali, O., Bate, A. & Flatz, L. Association Between Immune-Related Adverse Events During Anti–PD-1 Therapy and Tumor Mutational Burden. *JAMA Oncol.* (2019). doi:10.1001/jamaoncol.2019.3221

17. Yarchoan, M. et al. PD-L1 expression and tumor mutational burden are independent biomarkers in most cancers. *JCI Insight* 4, (2019).
18. Khoja, L., Day, D., Chen, T. W., Siu, L. L. & Hansen, A. R. Tumour- and class-specific patterns of immune-related adverse events of immune checkpoint inhibitors : a systematic review. *Ann. Oncol.* 2377–2385 (2017). doi:10.1093/annonc/mdx286
19. Oshima, Y., Tanimoto, T., Yuji, K. & Tojo, A. EGFR-TKI-associated interstitial pneumonitis in nivolumab-treated patients with non-small cell lung cancer. *JAMA Oncol.* 4, 1112–1115 (2018).
20. Berner, F. et al. Association of Checkpoint Inhibitor-Induced Toxic Effects with Shared Cancer and Tissue Antigens in Non-Small Cell Lung Cancer. *JAMA Oncol.* 1, 1043–1047 (2019).
21. Salem, J. E. et al. Cardiovascular toxicities associated with immune checkpoint inhibitors: an observational, retrospective, pharmacovigilance study. *Lancet Oncol.* 19, 1579–1589 (2018).
22. Wang, D. Y. et al. Fatal Toxic Effects Associated With Immune Checkpoint Inhibitors: A Systematic Review and Meta-analysis. *JAMA Oncol.* 4, 1721–1728 (2018).
23. Kirpich, A. et al. Variable selection in omics data: A practical evaluation of small sample sizes. *PLoS One* 13, 1–19 (2018).
24. Wu, C. & Ma, S. A selective review of robust variable selection with applications in bioinformatics. 16, 873–883 (2015).
25. Kwon, S., Oh, S. & Lee, Y. The use of random-effect models for high-dimensional variable selection problems. *Comput. Stat. Data Anal.* 103, 401–412 (2016).
26. Vasquez, M. M. et al. Least absolute shrinkage and selection operator type methods for the identification of serum biomarkers of overweight and obesity: simulation and application. *BMC Med. Res. Methodol.* 16, 1–19 (2016).
27. Zou, H. & Hastie, T. Regularization and variable selection via the elastic net. *J. R. Stat. Soc. Ser. B Stat. Methodol.* 67, 301–320 (2005).
28. Zhang, C. H. & Huang, J. The sparsity and bias of the lasso selection in high-dimensional linear regression. *Ann. Stat.* 36, 1567–1594 (2008).
29. Hayashi, T. et al. Visceral adiposity is an independent predictor of incident hypertension in Japanese Americans. *Ann. Intern. Med.* 140, (2004).

30. Gujarati, D. Multicollinearity: what happens if the regressors are correlated. in *Basic Econometrics* 320 (2009).
31. Kutner, M. H., Nachtsheim, C. J., Neter, J. & William Li. Multiple Regression II. in *Applied Linear Statistical Models* 283 (2005).

REVIEWER COMMENTS

Reviewer #1 (Remarks to the Author):

For the new Figure 3, I commend the authors for attempting answer this very important question, however, the data supplied are insufficient (or insufficiently presented, at the very least) to support their hypothesis. First, what type of tissue? The authors repeatedly report these are FFPE tissues stained for LCP1 and ADPGK. When (prediction requires sampling tissue before, not after or at toxicity) and where (are these pre-treatment tumors? How proximal to therapy if so? Are they from the tissue of toxicity of question, consistent with Supplemental Figure 9? - which has its own issues, see next point) are questions that remain and strangely was not reported anywhere that I could find. All of the other data are derived from tumors. In order for this to be important (particularly given that more inflamed tumors generally respond better and given the findings that irAEs are associated with response (as contentious as those data are) it remains the burden of proof to show that pre-treatment tumor staining with LCP1 and ADGPK, after controlling for CD8 or other immune-cell infiltration, predicts for irAEs.

For the mRNA validation in sites of toxicity (brain/encephalitis and cardiac myocarditis) - if LCP1 and ADPGK1 have immune or T cell-enriched expression, this is not useful given that in looking at the citations, smooth muscle and unaffected brain tissue were devoid of immune cells.

Reviewer #2 (Remarks to the Author):

None

Re: Manuscript ID: NCOMMS-20-10390A-Z

Reviewer #1 (Remarks to the Author):

For the new Figure 3, I commend the authors for attempting answer this very important question, however, the data supplied are insufficient (or insufficiently presented, at the very least) to support their hypothesis. First, what type of tissue? The authors repeatedly report these are FFPE tissues stained for LCP1 and ADPGK. When (prediction requires sampling tissue before, not after or at toxicity) and where (are these pre-treatment tumors?)

Response: Thanks for reviewer's suggestion, and we totally agree with the reviewer. We collected pre-treatment tumor tissues of cancer patients before receiving anti-PD-1/PD-L1 and any other treatment. We clarified this in our revised manuscript (Online Method section page 10, line 23 - page 11, line12; Result section page 6, line 13 & line 22).

How proximal to therapy if so?

Response: We collected pre-treatment tumor samples, so the timeline to therapy may not be relevant to our study. Nevertheless, we have added this time information, as well as other detailed information in Supplementary Table 9.

Are they from the tissue of toxicity of question, consistent with Supplemental Figure 9? - which has its own issues, see next point) are questions that remain and strangely was not reported anywhere that I could find.

Response: Data for Supplementary Figure 9 is from TCGA tumor samples, which is consistent with figure 3. We clarified this in the revised manuscript. We assumed that reviewer mean Supplementary Figure 8 with its own tissues. We agree with the reviewers that it is questionable to perform analysis in the tissues developed toxicities to predict irAEs (which we removed in the revised manuscript, please see our response below). Therefore, we collected pre-treatment tumor samples of patients before receiving anti-PD-1/PD-L1 and any other treatment to validate our conclusions (Online Method section page 10, line 23 – page 11, line 12).

All of the other data are derived from tumors. In order for this to be important (particularly given that more inflamed tumors generally respond better and given the findings that irAEs are associated with response (as contentious as those data are) it remains the burden of proof to show that pre-treatment tumor staining with LCP1 and ADGPK, after controlling for CD8 or other immune-cell infiltration, predicts for irAEs.

Response: All data is from tumor samples, except supplementary figure 8. We hypothesize that LCP1 and ADPGK might enable a pre-risk-check of patients before receiving anti-PD-1/PD-L1 agents. We hope that by checking the pre-treatment tumor sample at diagnosis, we will be able to predict the risk for irAE for this patient. For high risk patients, a close monitor is necessary when

treat with anti-PD-1/PD-L1 therapies.

We agree that irAE might associate with response, and we also explained this in our last response. We aimed to identify potential better biomarkers to predict irAEs, and we discovered that LCP1 ($R_s = 0.82$) and ADPGK ($R_s = 0.77$) performed better than CD8+ T cells ($R_s = 0.50$) and other immune cell infiltration, at least in our analytical setting.

To further address reviewer's concern about CD8 or other immune-cell infiltration, we performed CD8 IHC staining in the same pre-treatment tumor tissues we used for LCP1 and ADPGK staining. We did not observe significantly increased CD8 staining signals in pre-treatment tumors in irAE group (Figure A1). We also calculated the AUC value of CD8 and found the performance of CD8 is not comparable to LCP1 and ADPGK (Figure A1). Therefore, the confounding effect of CD8 should be very limited. We have added these results as Supplementary Fig. 13 and described in revised manuscript (page 7, line 5-9).

Figure A1. Determination of predictive power of CD8 in a patient cohort.

a. Quantification of CD8 IHC staining signals. Comparison between patients with irAE or without irAE is conducted by unpaired two-sided student's *t*-test.

b. ROC curve of CD8 in the validation cohort (n=28).

For the mRNA validation in sites of toxicity (brain/encephalitis and cardiac myocarditis) - if LCP1 and ADPGK1 have immune or T cell-enriched expression, this is not useful given that in looking at the citations, smooth muscle and unaffected brain tissue were devoid of immune cells.

Response: Thanks for reviewer's comment, we agree that this section is not useful and may cause confusion. we deleted this in the revised manuscript and Supplementary Figures.

Reviewer #3 also communicated to us that they thought that the number of samples used here was too few and that a diagram of how the tissues were selected would be welcomed.

Response: According to reviewer and editor's constructive suggestions, we drew the diagram to illustrate sample collection process of our validation cohort (Figure A2), and we also added it as Supplementary Figure 11 in our revised Supplementary Figures. We performed a retrospective review of cancer patients with lung cancer, gastrointestinal (GI) cancer, genitourinary (GU) cancer or other cancers receiving anti-PD-1/PD-L1 treatment from 2017 to 2019 in our hospital. Immune related adverse events (irAEs) are result from excessive immunity against normal organs induced by immune checkpoint blockade¹⁻³, and it is still challenge to determine irAE in clinical practice³⁻⁵. To identify high-confident irAE in patients, we used relatively stringent criteria: 1) we only include patients with CT confirmed pneumonitis; 2) we only include the pneumonitis that require and respond to steroids, immunosuppressants or endocrine therapies; 3) two investigators independently determine the pneumonitis as immunologic etiology (Figure A2). With this stringent criterion, only 39 out of 904 (4.3%) patients with clear pneumonitis induced by anti-PD-1/PD-L1 were selected for our validation. The percentage is consistent with previous studies^{2,6}, which is the key challenge to obtain large number of patient samples with irAEs. To be noticed, previously published high-impact irAE research papers only collected tissues from one or two irAE patients in their studies^{7,8}, and the sample size in our study is probably the largest, or one of the largest till date. Nevertheless, we discussed this limitation in our revised manuscript (page 8, line 19-20). Patients with incomplete demographic and follow-up information, receiving anti-PD-1/PD-L1 in other hospitals, or history of anti-CTLA-4 therapy were excluded. Next, we kept patients who have archived high-quality pre-treatment FFPE tumor samples before receiving anti-PD-1/PD-L1 and any other treatment. In addition, the tumor tissue should contain <30% necrosis. We collected comparable number of anti-PD-1/PD-L1 treated cancer patients without any overserved irAEs with matched cancer types, stage, age, sex, and therapy of irAE group. We clarified this in details in the online methods section in our revised manuscript (page 10, line 23 – page 11, line 12).

Figure A2. Diagram for the sample collection of validation cohort.

Gastrointestinal (GI) cancer include gastric, liver, colorectal and gallbladder cancer; Genitourinary (GU) cancer include kidney, bladder, prostate cancer.

References

1. Esfahani, K. et al. Moving towards personalized treatments of immune-related adverse events. *Nat. Rev. Clin. Oncol.* (2020) doi:10.1038/s41571-020-0352-8.
2. Kennedy, L. B. & Salama, A. K. S. A review of cancer immunotherapy toxicity. *CA. Cancer J. Clin.* (2020) doi:10.3322/caac.21596.
3. Ramos-Casals, M. et al. Immune-related adverse events of checkpoint inhibitors. *Nature Reviews Disease Primers* vol. 6 1–21 (2020).
4. Zhu, A. X. et al. Pembrolizumab in patients with advanced hepatocellular carcinoma previously treated with sorafenib (KEYNOTE-224): a non-randomised, open-label phase 2 trial. *Lancet Oncol.* 19, 940–952 (2018).
5. Kaufman, H. L. et al. Avelumab in patients with chemotherapy-refractory metastatic Merkel cell carcinoma: a multicentre, single-group, open-label, phase 2 trial. *Lancet Oncol.* 17, 1374–1385 (2016).
6. Martins, F. et al. Adverse effects of immune-checkpoint inhibitors: epidemiology, management and surveillance. *Nat. Rev. Clin. Oncol.* 16, 563–80 (2019).
7. Johnson, D. B. et al. A case report of clonal EBV-like memory CD4+ T cell activation in fatal checkpoint inhibitor-induced encephalitis. *Nat. Med.* 25, 1243–1250 (2019).
8. Johnson, D. B. et al. Fulminant myocarditis with combination immune checkpoint blockade. *N. Engl. J. Med.* 375, 1749–1755 (2016).